# Unsupervised Learning of Neurosymbolic Encoders

**Eric Zhan\***
*California Institute of Technology*

*ezhan@caltech.edu*

**Jennifer J. Sun\***
*California Institute of Technology*

*jjsun@caltech.edu*

**Ann Kennedy**
*Northwestern University Feinberg School of Medicine*

**Yisong Yue**
*California Institute of Technology*

**Swarat Chaudhuri**
*University of Texas at Austin*

**\* Equal contribution**

**Reviewed on OpenReview: `https://openreview.net/forum?id=eWvBEMTlRq`**

## Abstract

We present a framework for the unsupervised learning of neurosymbolic encoders, which are encoders obtained by composing neural networks with symbolic programs from a domain-specific language. Our framework naturally incorporates symbolic expert knowledge into the learning process, which leads to more interpretable and factorized latent representations compared to fully neural encoders. We integrate modern program synthesis techniques with the variational autoencoding (VAE) framework, in order to learn a neurosymbolic encoder in conjunction with a standard decoder. The programmatic descriptions from our encoders can benefit many analysis workflows, such as in behavior modeling where interpreting agent actions and movements is important. We evaluate our method on learning latent representations for real-world trajectory data from animal biology and sports analytics. We show that our approach offers significantly better separation of meaningful categories than standard VAEs and leads to practical gains on downstream analysis tasks, such as for behavior classification. Code can be found at `https://github.com/ezhan94/neurosymbolic-encoders`.

## 1 Introduction

Advances in unsupervised learning have enabled the discovery of latent structures in data from a variety of domains, such as image data (Dupont, 2018), sound recordings (Calhoun et al., 2019), and tracking data (Luxem et al., 2020). For instance, a common approach is to use encoder-decoder frameworks, such as variational autoencoders (VAEs) (Kingma & Welling, 2014), to identify a low-dimensional latent representation from the raw data that could contain disentangled factors of variation (Dupont, 2018) or semantically meaningful clusters (Luxem et al., 2020). Such approaches typically employ complex mappings based on neural networks, and explaining how the model assigns inputs to latent representations can be challenging (Zhang et al., 2020).

In this paper, we introduce *unsupervised neurosymbolic representation learning*, which allows part of a representation to be computed using symbolic *encoder programs* written in a predefined domain-specific language (DSL). (The rest of the representation is computed using a neural network.) The use of such

neurosymbolic encoders can offer two key benefits over purely neural approach. First, since a DSL reflects structured domain knowledge, neurosymbolic encoders can often produce representations that are human-interpretable (Verma et al., 2018; Shah et al., 2020). Second, as observed in studies that used hand-crafted programmatic encoders (Zhan et al., 2020), these representations can potentially be more factorized or well-separated into meaningful categories than purely neural representations.

Our learning algorithm is grounded in the VAE framework (Kingma & Welling, 2014; Mnih & Gregor, 2014) and aims to discover a neurosymbolic encoder coupled with a standard neural decoder.[1] A key challenge here is that the space of programs in a DSL is combinatorial. We tackle this problem by assuming programs to be differentiable and by tightly integrating standard VAE training with modern program synthesis methods (Chaudhuri et al., 2021; Shah et al., 2020). We further show how to incorporate ideas from adversarial information factorization (Creswell et al., 2017) and enforcing capacity constraints (Burgess et al., 2017; Dupont, 2018) in order to mitigate issues such as posterior and index collapse in the learned representation.

Programmatic descriptions from neurosymbolic encoders are especially useful in behavior analysis (Segalin et al., 2020; Sun et al., 2021b), where domain experts routinely interpret clusters of behaviors as part of an analysis workflow. Accordingly, our experimental evaluation focuses on this setting. By integrating domain knowledge using program synthesis, we demonstrate that our clusters are inherently interpretable and better aligned with human-annotated labels across multiple behavior analysis datasets. To validate the end-to-end practicality for analysis workflows, we integrate our automatically learned programs into a state-of-the-art behavior analysis framework, Task Programming (Sun et al., 2021b), that typically relies on expert-crafted programs, and demonstrate competitive performance using our automatically synthesized programs.

To summarize, our contributions are:

- We propose a neurosymbolic approach to representation learning, in which part of the latent representation is produced by an interpretable encoder program, while the rest is computed using a neural network.

- We realize the approach via a learning algorithm that combines VAE training and program synthesis.

- We show that our approach can significantly outperform purely neural encoders in extracting semantically meaningful representations of behavior, as measured by standard unsupervised metrics.

- We further explore the flexibility of our approach, by showing that performance can be robust across different DSL designs by domain experts.

- We showcase the practicality of our approach on downstream tasks, by incorporating our approach into a state-of-the-art self-supervised learning approach for behavior analysis (Sun et al., 2021b).

## 2 Background

### 2.1 Variational Autoencoders

We build on VAEs (Kingma & Welling, 2014; Mnih & Gregor, 2014), a latent variable modeling framework shown to learn effective latent representations (also called encodings/embeddings) (Higgins et al., 2016; Zhao et al., 2017; Yingzhen & Mandt, 2018) and can capture the generative process (Oord et al., 2017; Vahdat & Kautz, 2020; Zhan et al., 2020). VAEs introduce a latent variable $\mathbf{z}$, an encoder $q_\phi$, a decoder $p_\theta$, and a prior distribution $p$ on $\mathbf{z}$. $\phi$ and $\theta$ are the parameters of the $q$ and $p$ respectively, often instantiated with neural networks. The learning objective is to maximize the evidence lower bound (ELBO) of the data log-likelihood:

$$\text{ELBO} := \mathbb{E}_{q_\phi(\mathbf{z}|\mathbf{x})} \left[ \log p_\theta(\mathbf{x}|\mathbf{z}) \right] - D_{KL}\big(q_\phi(\mathbf{z}|\mathbf{x})||p(\mathbf{z})\big) \leq \log p(\mathbf{x}). \tag{1}$$

The first term in Eq. 1 is the log-density assigned to the data, while the second term is the KL-divergence between the prior and approximate posterior of $\mathbf{z}$. Latent representations $\mathbf{z}$ are often continuous and modeled with a Gaussian prior, but $\mathbf{z}$ can be modeled to contain discrete dimensions as well (Kingma et al., 2014;

---

[1]Some prior work have studied the complementary problem of learning (neuro-)symbolic decoders (e.g,. Ellis et al. (2018); Feinman & Lake (2020)). See Section 5 for more discussion.

Hu et al., 2017; Dupont, 2018). Our experiments are focused on behavioral tracking data in the form of trajectories, and so in practice we utilize a trajectory variant of VAEs (Co-Reyes et al., 2018; Zhan et al., 2020; Sun et al., 2021b), described in Section 3.5.

One challenge with VAEs (and deep encoder-decoder models in general) is that while the model is expressive, it is often difficult to interpret what is encoded in the latent representation $\mathbf{z}$. Common approaches include taking traversals in the latent space and visualizing the resulting generations (Burgess et al., 2017), or post-processing the latent variables using techniques such as clustering (Luxem et al., 2020). Such techniques are post-hoc and thus cannot guide (in an interpretable way) the encoder to be biased towards a family of structures. Some recent work have studied how to impose structure in the form of graphical models or dynamics in the latent space (Johnson et al., 2016; Deng et al., 2017), and our work can be thought of as a first step towards imposing structure in the form of symbolic knowledge encoded in a domain specific programming language.

## 2.2 Synthesis of Differentiable Programs

Our approach utilizes recent work on the synthesis of differentiable programs (Chaudhuri et al., 2021; Shah et al., 2020; Valkov et al., 2018), where one learns both the discrete structure of the symbolic program (analogous to the architecture of a neural network) as well as the differentiable parameters within that structure. Our formulation closely follows that of Shah et al. (2020). We use a domain-specific programming language (DSL), generated with a context-free grammar (see Figure 3 for an example). A program is represented as a pair $(\alpha, \psi)$, where $\alpha$ is a discrete program architecture and $\psi$ are its real-valued parameters. We denote $\mathcal{P}$ as the space of symbolic programs (i.e. programs with complete architectures). The semantics of a program $(\alpha, \psi)$ is given by a function $[\![\alpha]\!](x, \psi)$ that is guaranteed to be differentiable in both $x$ and $\psi$.

Like Shah et al. (2020), we pose the problem of learning differentiable programs as search through a directed program graph $\mathcal{G}$. The graph $\mathcal{G}$ models the top-down construction of program architectures $\alpha$ through the repeated firing of rules of the DSL grammar, starting with an *empty* architecture $\alpha_0$ (represented by the "start" nonterminal of the grammar). The *leaf nodes* of $\mathcal{G}$ represent programs with *complete* architectures (no nonterminals). Thus, $\mathcal{P}$ is the set of programs in the leaf nodes of $\mathcal{G}$. The other nodes in $\mathcal{G}$ contain programs with *partial* architectures (has at least one nonterminal). We interpret a program in a non-leaf node as being neurosymbolic, by viewing its nonterminals as representing neural networks with free parameters. The root node in $\mathcal{G}$ is the empty architecture $\alpha_0$, interpreted as a fully neural program. An edge $(\alpha, \alpha')$ exists in $\mathcal{G}$ if one can obtain $\alpha'$ from $\alpha$ by applying a rule in the DSL that replaces a nonterminal in $\alpha$.

Program synthesis in this problem setting equates to searching through $\mathcal{G}$ to find the optimal complete program architecture, and then learning corresponding parameters $\psi$, i.e., to find the optimal $(\alpha, \psi)$ that minimizes a combination of standard training loss (e.g., classification error) and structural loss (preferring "simpler" $\alpha$'s). Shah et al. (2020) evaluate multiple strategies for solving this problem and finds *informed search using admissible neural heuristics* to be the most efficient strategy (see appendix). Consequently, we adopt this algorithm for our program synthesis task.

## 3 Neurosymbolic Encoders

The structure of our neurosymbolic encoder is shown in the right diagram of Figure 1. The latent representation $\mathbf{z} = [\mathbf{z}_\phi, \mathbf{z}_{(\alpha, \psi)}]$ is partitioned into neurally encoded $\mathbf{z}_\phi$ and programmatically encoded $\mathbf{z}_{(\alpha, \psi)}$. This approach boasts several advantages:

- The symbolic component of the latent representation is programmatically interpretable.

- The neural component can encode any residual information not captured by the program, which maintains the model's capacity compared to deep encoders (see synthetic experiment in Section 4.2).

- By incorporating a modular design, we can leverage state-of-the-art learning algorithms for both differentiable encoder-decoder training and program synthesis.

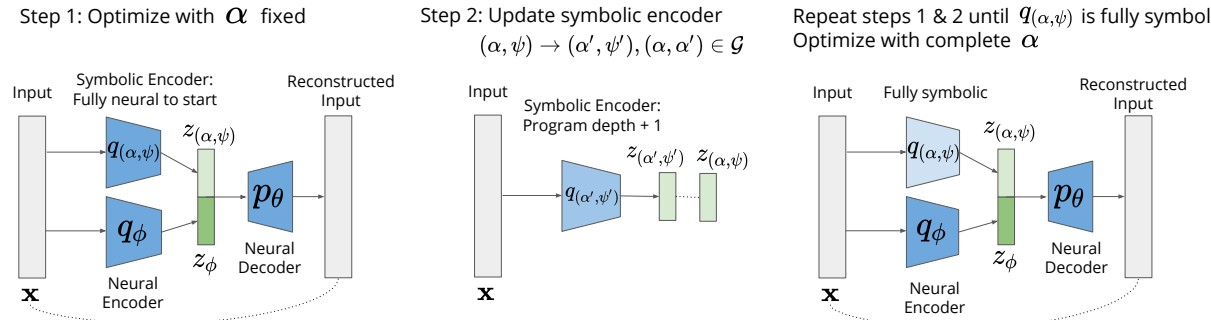

Figure 1: **Learning Neurosymbolic Encoders: Sketch of Algorithm 1** (Section 3.1). The symbolic encoder is initially fully neural. We alternate between VAE training with the program architecture fixed (Step 1 as in Eq. 2), and supervised program learning to increase the depth of the program by 1 (Step 2 as in Eq. 3). Once we reach a symbolic program, we train the model one last time to learn all the parameters. The color (in terms of lightness) of the symbolic encoder corresponds to the encoder becoming more symbolic over time.

We denote $q_\phi$ and $q_{(\alpha,\psi)}$ as the neural and symbolic encoders respectively (see Figure 1), where $\mathbf{z}_\phi \sim q_\phi(\cdot|\mathbf{x})$ and $\mathbf{z}_{(\alpha,\psi)} \sim q_{(\alpha,\psi)}(\cdot|\mathbf{x})$. $q_\phi$ is instantiated with a neural network, but $q_{(\alpha,\psi)}$ is a differentiable program with architecture $\alpha$ and parameters $\psi$ in some program space $\mathcal{P}$ defined by a DSL. Given an unlabeled training set of $\mathbf{x}$'s, our neurosymbolic-VAE (ns-vae) learning objective becomes:

$$
\max_{\phi,(\alpha,\psi),\theta} \quad \mathcal{L}_{\text{ns-vae}}(\phi, \alpha, \psi, \theta)
$$
$$
= \max_{\phi,(\alpha,\psi),\theta} \quad \mathbb{E}_{q_\phi(\mathbf{z}_\phi|\mathbf{x})q_{(\alpha,\psi)}(\mathbf{z}_{(\alpha,\psi)}|\mathbf{x})} \Big[ \underbrace{\log p_\theta(\mathbf{x}|\mathbf{z}_\phi, \mathbf{z}_{(\alpha,\psi)})}_{\text{reconstruction loss}} \Big]
$$
$$
- \underbrace{D_{KL}\big(q_\phi(\mathbf{z}_\phi|\mathbf{x})||p(\mathbf{z}_\phi)\big)}_{\text{regularization for neural latent}} - \underbrace{D_{KL}\big(q_{(\alpha,\psi)}(\mathbf{z}_{(\alpha,\psi)}|\mathbf{x})||p(\mathbf{z}_{(\alpha,\psi)})\big)}_{\text{regularization for symbolic latent}}.
$$

(2)

Compared to the standard VAE objective in Eq. 1 for a single neural encoder, Eq. 2 has separate KL-divergence terms for the neural and programmatic encoders.

## 3.1 Learning Algorithm

The challenge with solving for Eq. 2 is that while $(\phi, \psi, \theta)$ can be optimized via back-propagation with $\alpha$ fixed, optimizing for $\alpha$ is a discrete optimization problem. Since it is difficult to jointly optimize over both continuous and discrete spaces, we take an iterative, alternating optimization approach. We start with a fully neural program (one with empty architecture $\alpha_0$ as described in Section 2.2) trained using standard differentiable optimization (Figure 1, Step 1). We then gradually make it more symbolic (Figure 1, Step 2) by finding a program that is a child of the current program in $\mathcal{G}$ (more symbolic by construction of $\mathcal{G}$) that outputs as similar to the current latent representations as possible:

$$
\min_{\alpha':(\alpha,\alpha')\in\mathcal{G},\ \psi'} \quad \mathcal{L}_{\text{supervised}}\big(q_{(\alpha,\psi)}(\mathbf{x}), q_{(\alpha',\psi')}(\mathbf{x})\big),
$$

(3)

which can be viewed as a form of distillation (from less symbolic to more symbolic programs) via matching the input/output behavior. We solve Eq. 3 by enumerating over all child programs of the current search tree and selecting the best one, which is similar to one iteration of iteratively-deepened depth-first search in Shah et al. (2020) (more details in Section 3.2). We alternate between optimizing Eq. 2 and Eq. 3 until we obtain a complete program. Algorithm 1 outlines this procedure and is guaranteed to terminate if $\mathcal{G}$ is finite by specifying a maximum program depth.

We chose this optimization procedure for two reasons. First, it maximally leverages state-of-the-art tools in both differentiable latent variable modeling (VAE-style training) and supervised program synthesis (for distillation), leading to tractable algorithm design. Second, this procedure never makes a drastic change to the program architecture, leading to relatively stable learning behavior across iterations.

**Algorithm 1** Learning a neurosymbolic encoder

1: **Input**: program space $\mathcal{P}$, program graph $\mathcal{G}$
2: initialize $\phi, \psi, \theta, \alpha = \alpha_0$ (empty architecture)
3: **while** $\alpha$ is not complete **do**
4:      $\phi, \psi, \theta \leftarrow$ optimize Eq. 2 with $\alpha$ fixed
5:      $(\alpha, \psi) \leftarrow$ optimize Eq. 3
6: **end while**
7: $\phi, \psi, \theta \leftarrow$ optimize Eq. 2 with complete $\alpha$
8: **Return**: encoder $\{q_\phi, q_{(\alpha, \psi)}\}$

**Algorithm 2** Learning a neurosymbolic encoder with $k$ programs

1: **Input**: program space $\mathcal{P}$, program graph $\mathcal{G}$, $k$
2: **for** $i = 1..k$ **do**
3:      fix programs $\{q_{(\alpha_1, \psi_1)}, \ldots, q_{(\alpha_{i-1}, \psi_{i-1})}\}$
4:      execute Algorithm 1 to learn $q_{(\alpha_i, \psi_i)}$
5:      remove $q_{(\alpha_i, \psi_i)}$ from $\mathcal{P}$ to avoid redundancies
6: **end for**
7: **Return**: encoder $\{q_\phi, q_{(\alpha_1, \psi_1)}, \ldots, q_{(\alpha_k, \psi_k)}\}$

### 3.2 Program Synthesis via NEAR

Our strategy for solving Eq. 3 utilizes the setup in Shah et al. (2020). We summarize the key points below.

**Program graph $\mathcal{G}$.** Shah et al. (2020) learns programs in a supervised learning setting that minimizes a structural cost $s$ (deeper programs are more costly) and a prediction error $\zeta$:

$$(\alpha^*, \psi^*) = \underset{(\alpha, \psi)}{\arg \min}(s(\alpha) + \zeta(\alpha, \psi)). \tag{4}$$

Shah et al. (2020) construct a program graph $\mathcal{G}$ such that solving Eq. 4 equates to finding a leaf node with the minimum path cost on $\mathcal{G}$. We include a copy of their illustration for $\mathcal{G}$ in Figure 2. Our problem definition in Eq. 3 is very similar so we utilize the same program graph. The difference is that our labels are not ground-truth but rather the labels assigned by the current neurosymbolic encoder.

**Neural heuristic $h$.** Shah et al. (2020) solve Eq. 4 by introducing a heuristic as a neural admissible relaxation (NEAR for short). Leveraging a fully differentiable DSL, they use neural networks to fill in for nonterminals in programs and show that the performance of such neurosymbolic programs are underestimates of the total path costs of descendent leaf nodes and thus, can be used as an admissible heuristic. This allows them to integrate their heuristic with several graph search algorithms, of which they adopt A* search and iteratively-deepened depth-first-search with branch-and-bound (IDS-BB). We use IDS-BB in our work.

**IDS-BB.** The full algorithm for IDS-BB is described in Algorithm 2 in Shah et al. (2020). In our work, this reduces to the following:

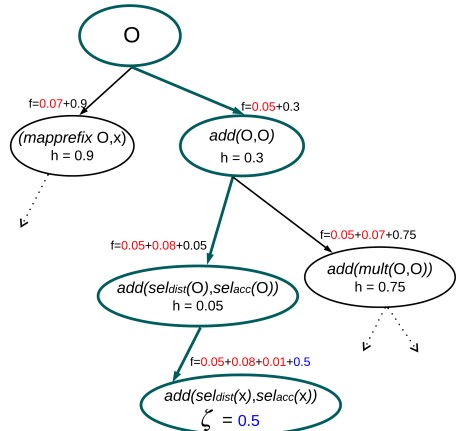

1. For the current program, we enumerate its children in $\mathcal{G}$.

2. We compute the heuristic for each child in $\mathcal{G}$ by replacing any nonterminals with neural networks.

3. We commit to the most promising child (with respect to the heuristic) and update the program, which can be viewed as one iteration of the full IDS-BB algorithm.

Figure 2: Figure 4 from Shah et al. (2020). Structural costs $s$ are in red, heuristic values $h$ in black, and prediction errors $\zeta$ in blue. O refers to nonterminals.

One key difference is that the original IDS-BB algorithm maintains a frontier ordered by the best heuristics encountered so far. However, our label distributions can change between iterations (since the symbolic component of the encoder is updated and thus, so are the labels it assigns), which invalidates the heuristics computed from previous iterations. This leaves a very interesting direction for future work.

### 3.3 Learning Multiple Programs

The interpretability of latent representations induced by symbolic encoders $q_{(\alpha, \psi)}$ ultimately depends on the DSL. For instance, a program that encodes to one of ten classes may not be very interpretable if it involves a

matrix multiplication within the program. Instead, we learn *binary* programs that encode sequences into one of two classes (using binary cross-entropy for $\mathcal{L}_{\text{supervised}}$, a uniform prior on 2-dimensional $\mathbf{z}_{(\alpha,\psi)}$, and Gumbel-Softmax (Jang et al., 2017) to sample $\mathbf{z}_{(\alpha,\psi)}$ from the posterior). Figures 5a & 5b depict learned binary programs that encode mice trajectories and their interpretations.

To encode more than two classes, we simply learn multiple binary programs by extending Eq. 2 to sum over $\mathcal{L}_{\text{supervised}}$ for $k$ symbolic programs $\{q_{(\alpha_1,\psi_1)}, \ldots, q_{(\alpha_k,\psi_k)}\}$ and corresponding latent representations $\{\mathbf{z}_{(\alpha_1,\psi_1)}, \ldots, \mathbf{z}_{(\alpha_k,\psi_k)}\}$. This results in $2^k$ classes and a solution space that now scales exponentially (e.g. $|\mathcal{P}|^k$). Algorithm 2 outlines our greedy solution that reuses Algorithm 1 by iteratively learning one symbolic program at a time. We leave the exploration of more sophisticated search methods as future work.

### 3.4 Dealing with Posterior and Index Collapse

Deep latent variable models, especially those with discrete latent variables, are notoriously prone to both posterior (Bowman et al., 2015; Chen et al., 2016b; Oord et al., 2017) and index (Kaiser et al., 2018) collapse. Since our algorithms optimize for such models repeatedly, they can be susceptible to these failure modes. Below, we summarize two strategies that we found to work well in our setting.[2]

**Adversarial information factorization.** Index collapse is the phenomenon in which all data is encoded into one class, resulting in a discrete latent variable $\mathbf{z}_{(\alpha,\psi)}$ that is effectively meaningless. Creswell et al. (2017) counteracts index collapse by introducing an adversarial network $A_\omega$ and maximizing the adversarial loss below to ensure that the adversary $A_\omega$ cannot successfully predict $\mathbf{z}_{(\alpha,\psi)}$ from $\mathbf{z}_\phi$.

$$
\begin{aligned}
&\max_{\phi,(\alpha,\psi),\theta} \quad \mathcal{L}_{\text{fac}}(\phi,\alpha,\psi,\theta) \\
&= \max_{\phi,(\alpha,\psi),\theta} \quad \mathbb{E}_{q_\phi(\mathbf{z}_\phi|\mathbf{x})q_{(\alpha,\psi)}(\mathbf{z}_{(\alpha,\psi)}|\mathbf{x})} \Big[ \log p_\theta(\mathbf{x}|\mathbf{z}_\phi,\mathbf{z}_{(\alpha,\psi)}) + \underbrace{\min_\omega \mathcal{L}_{\text{adv}}\big(A_\omega(\mathbf{z}_\phi),\mathbf{z}_{(\alpha,\psi)}\big)}_{\text{adversary}} \Big] \\
&\qquad\qquad - D_{KL}\big(q_\phi(\mathbf{z}_\phi|\mathbf{x})||p(\mathbf{z}_\phi)\big) - D_{KL}\big(q_{(\alpha,\psi)}(\mathbf{z}_{(\alpha,\psi)}|\mathbf{x})||p(\mathbf{z}_{(\alpha,\psi)})\big)
\end{aligned}
\tag{5}
$$

**Channel capacity constraint.** Posterior collapse is the phenomenon in which the posterior trivially matches the prior exactly (a KL-divergence of 0) but the latent variables are unused by the decoder. Burgess et al. (2017) and Dupont (2018) instead force the KL-divergence terms to match capacities $C_\phi$ and $C_{(\alpha,\psi)}$, which are hyperparameter (see appendix). Since the KL-divergence is an upper bound on the mutual information between latent variables and the data (Kim & Mnih, 2018; Dupont, 2018), this encourages the latent variables to encode information and aims to prevent posterior collapse.

$$
\begin{aligned}
&\max_{\phi,(\alpha,\psi),\theta} \quad \mathcal{L}_{\text{cap}}(\phi,\alpha,\psi,\theta) \\
&= \max_{\phi,(\alpha,\psi),\theta} \quad \mathbb{E}_{q_\phi(\mathbf{z}_\phi|\mathbf{x})q_{(\alpha,\psi)}(\mathbf{z}_{(\alpha,\psi)}|\mathbf{x})} \Big[ \log p_\theta(\mathbf{x}|\mathbf{z}_\phi,\mathbf{z}_{(\alpha,\psi)}) \Big] \\
&\qquad - \gamma_\phi |D_{KL}\big(q_\phi(\mathbf{z}_\phi|\mathbf{x})||p(\mathbf{z}_\phi)\big) - C_\phi| - \gamma_{(\alpha,\psi)} |D_{KL}\big(q_{(\alpha,\psi)}(\mathbf{z}_{(\alpha,\psi)}|\mathbf{x})||p(\mathbf{z}_{(\alpha,\psi)})\big) - C_{(\alpha,\psi)}|
\end{aligned}
\tag{6}
$$

In our algorithms, we augment our initial objective in Eq. 2 with Eq. 5 and Eq. 6:

$$
\max_{\phi,(\alpha,\psi),\theta} \mathcal{L}_{\text{ns-vae}}(\phi,\alpha,\psi,\theta) + \lambda_{\text{fac}}\mathcal{L}_{\text{fac}}(\phi,\alpha,\psi,\theta) + \lambda_{\text{cap}}\mathcal{L}_{\text{cap}}(\phi,\alpha,\psi,\theta),
\tag{7}
$$

where $\lambda_{\text{fac}} = \lambda_{\text{cap}} = 1$ in our experiments.

### 3.5 Instantiation for Sequential Domains

The objective in Eq. 2 describes a general problem that is applicable to any domain. In our experiments, we focus on sequential trajectory data. Trajectory data is often used in scientific applications where interpretability is desirable, such as behavior discovery (Luxem et al., 2020; Hsu & Yttri, 2020). The ability to easily explain the learned latent representation using programs can help domain experts better understand the structure in their data. Trajectory data is also often relatively low dimensional, which helps experts encode domain knowledge into the DSL more easily (Shah et al., 2020; Sun et al., 2021b; Zhan et al., 2020).

---

[2]There are many approaches available for tackling both these issues, but we emphasize that these contributions are orthogonal to ours; as techniques for preventing posterior and index collapse improve, so will the robustness of our algorithm.

$$\alpha \quad ::= \quad x \mid \oplus(\alpha_1, \ldots, \alpha_k) \mid \oplus_\theta(\alpha_1, \ldots, \alpha_k)$$
$$\textbf{if } \alpha_1 \textbf{ then } \alpha_2 \textbf{ else } \alpha_3 \mid \textbf{sel}_S \ x \mid \textbf{mapaverage } (\textbf{fun } x_1.\alpha_1) \ x$$

Figure 3: Our DSL for sequential domains, similar to the one used in Shah et al. (2020). $x$, $\oplus$, and $\oplus_\theta$ represent inputs, basic algebraic operations, and parameterized library functions, respectively. **fun** $x.e(x)$ represents a function that evaluates an expression $e(x)$ over the input $x$. **sel**$_S$ selects a subset $S$ of the dimensions of the input $x$. **mapaverage** $g \ x$ applies the function $g$ to every element of the sequence $x$ and returns the average of the results. We employ a differentiable approximation of the **if-then-else** construct.

In this domain, $\mathbf{x}$ is a trajectory of length $T$: $\mathbf{x} = \{x_1, \ldots, x_T\}$. We then factorize the log-density in Eq. 2 as a product of conditional probabilities:

$$\log p_\theta(\mathbf{x}|\mathbf{z}_\phi, \mathbf{z}_{(\alpha,\psi)}) = \sum_{t=1}^{T} \log p_\theta(x_t|x_{<t}, \mathbf{z}_\phi, \mathbf{z}_{(\alpha,\psi)}). \tag{8}$$

When $q_\phi$ and $p_\theta$ are instantiated with recurrent neural networks (RNN), the model is more commonly known as a trajectory-VAE (TVAE) (Co-Reyes et al., 2018).

As the symbolic encoder $q_{(\alpha,\psi)}$ maps sequences to vectors, we adopt a DSL (Figure 3) previously used for sequence classification (Shah et al., 2020). Our DSL is purely functional and contains both basic algebraic operations and parameterized library functions. Domain experts can easily augment the DSL with their own functions, such as selection functions that select subsets of features that they deem potentially important. We ensure that all programs in our DSL are differentiable by utilizing a smooth approximation of the **if-then-else** construct (Shah et al., 2020). Figures 5a & 5b depict example programs (full details in the appendix).

## 4 Experiments

We take a multi-faceted approach to evaluate our unsupervised learning approach using synthetic data and real-world data from animal behavior and sports analytics. We also show the end-to-end practicality of our programs by applying them to a downstream behavior classification framework. Our research questions are:

- **Q1: Are the clusters created with our programs meaningful?** (Section 4.2). We evaluate this aspect both qualitatively and quantitatively by comparing with the truth generative process on synthetic datasets, as well as by comparing to human annotated labels on real-world datasets.

- **Q2: How sensitive is our approach to different DSL choices?** (Section 4.3). We compare programs learned in our framework from three different DSLs designed by three domain experts for studying animal behavior. The three DSLs (DSL 1, DSL 2, DSL 3) mainly differ in the behavioral features chosen by experts, and are described in Appendix Section C.

- **Q3: Are the programs useful for downstream tasks?** (Section 4.4). Ultimately, the practicality of these methods must be validated by their usefulness in downstream tasks such as those used in scientific analyses. We apply our unsupervised programs to a behavior classification framework called task programming (Sun et al., 2021b). This framework uses hand-crafted programs for self-supervision, which we replace with our automatically learned programs.

### 4.1 Experimental Setup

#### 4.1.1 Datasets

We summarize the datasets used in our experiments, and provide full details in the appendix.

**Synthetic.** We generate synthetic trajectories by sampling initial positions and velocities from a Gaussian distribution and introducing 2 ground-truth factors of variation as large external forces in the positive/negative x/y directions that affect velocity, totaling to 4 discrete classes. Velocities are sampled and fixed for the entire trajectory, but we also sample small Gaussian noise at each timestep. We generate 10k/2k/2k trajectories of length 25 for train/validation/test. Figure 4a shows 50 trajectories from the training set. This dataset

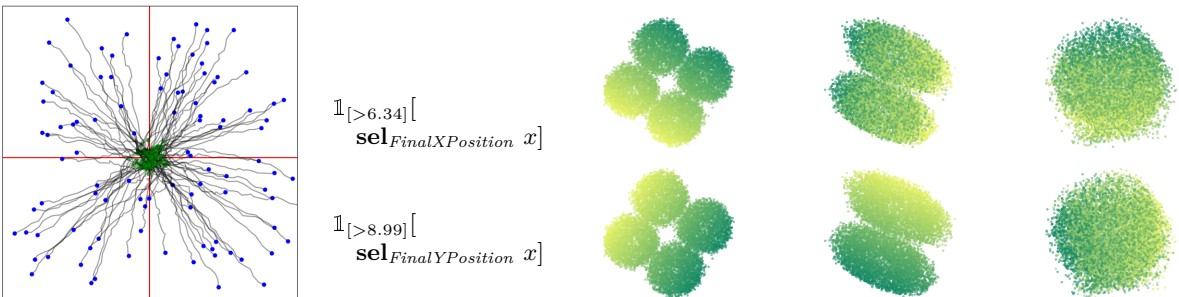

(a) 50 synthetic trajectories    (b) learned programs   (c) $\mathbf{z}_\phi$, 0 programs    (d) $\mathbf{z}_\phi$, 1 program    (e) $\mathbf{z}_\phi$, 2 programs

Figure 4: **Synthetic dataset experiments.** **(a)** Trajectories in synthetic training set. Initial/final positions are indicated in green/blue. Red lines delineate ground-truth classes, based on final positions. **(b)** $k = 2$ learned binary programs using our algorithm. The first program (top) thresholds the final x-position while the second program (bottom) thresholds the final y-position. **(c, d, e)** Neural latent variables reduced to 2 dimensions. Top/bottom rows are colored by final x/y-positions respectively (green/yellow is positive/negative). **(c)** Clusters in TVAE neural latent space correspond to 4 ground-truth classes. **(d)** After learning the first program, the neural latent space contains clusters only based on the final y-position. **(e)** After learning the second program, all 4 ground-truth classes have been extracted as programs and the remaining neural latent space contains no clear clustering.

is useful because we can evaluate whether our algorithm can learn programs that match the ground-truth factors of variation (such ground-truth information is not available in real-world datasets).

**CalMS21.** Our primary real-world dataset is the CalMS21 dataset (Sun et al., 2021a), containing trajectories of socially interacting mice captured for neuroscience experiments. Each frame contains 7 tracked keypoints for each of two mice. The dataset has one set of unlabeled tracking data, which we use to train our neurosymbolic encoder, and another set annotated with 4 labels at each frame by human experts (frame-level behaviors), which we use to evaluate our programs. These labels consists of three behaviors-of-interest between mice (attack, mount, investigation), and a label corresponding to all other behaviors (other), with a more detailed description in Sun et al. (2021a). Specifically, our evaluation uses labels from the test split of the CalMS21 classification task. We have 231k/52k/262k trajectories of length 21 for train/val/test. The features in our DSL are selected by a domain expert based on the attributes from Segalin et al. (2020).

**Basketball.** We use the same basketball dataset as in Shah et al. (2020) and Zhan et al. (2020) that tracks professional basketball players. Each trajectory is of length 25 over 8 seconds and contains the $xy$-positions of 10 players. We split trajectories by grouping offensive and defensive players (5 each), effectively doubling the dataset size. We evaluate our algorithm and the baselines with respect to the labels of offensive/defensive players. Our DSL includes additional domain features like player speed and distance-to-basket. In total, we have 177k/31k/27k trajectories for train/val/test.

### 4.1.2 Quantitative Evaluation Setup

The quantitative evaluations are used to compare our neurosymbolic encoders with baseline unsupervised learning methods on the real-world datasets.

**Baselines.** We compare our model containing a neurosymbolic encoder against other approaches based on VAEs. In particular, we compare against JointVAE (Dupont, 2018), which also has both discrete and continuous latent representations, and can be viewed as a fully neural version of our neurosymbolic encoder. Other baselines include VAE, VAE with K-means loss (Ma et al., 2019; Luxem et al., 2020), and Beta-VAE (Burgess et al., 2017). These models have a fully neural encoder and learn continuous latent representations, which we can then use to produce clusters with K-means clustering (Lloyd, 1982). We additionally compare against VQ-VAEs (Oord et al., 2017), which produce discrete latent clusters. We use the TVAE version of all baselines (details included in the appendix).

**Metrics.** Unlike in the synthetic setting, we do not have ground truth programs in the real-world datasets. We thus evaluate our programs quantitatively using (1) standard cluster metrics relative to human-defined

$$\mathbb{1}_{[>-7]} \begin{bmatrix} \textbf{mapaverage } (\textbf{fun } x_t. \\ \quad \textbf{multiply } (ResidentSpeedAffine_{[-6.3];-8.3}(x_t), \\ \quad NoseTailDistAffine_{[.04];-9.1}(x_t)) \ x \end{bmatrix}$$

(a) **Program learned using CalMS21 DSL 1**, resulting NMI 0.428. Since speed is positive, the first term is always negative. One cluster thus generally consists of trajectories where the mice are further apart, such that the second term is positive, and the negative product is less than the threshold. The other cluster generally occurs when the mice are close together, the second term is negative, and the product will be positive.

$$\mathbb{1}_{[>-5.7]} \begin{bmatrix} \textbf{mapaverage } (\textbf{fun } x_t. \\ \quad \textbf{add } (ResidentAxisRatioAffine_{[-8.0];-7.1}(x_t), \\ \quad BoundingBoxIOUAffine_{[-16.6];5.9}(x_t)) \ x \end{bmatrix}$$

(b) **Program learned using CalMS21 DSL 2**, resulting NMI 0.320. The axis ratio is the ratio of major axis length and minor axis length of an ellipse fitted to the mouse keypoints. The second term measures the bounding box overlap between mice, and is zero when the mice are far apart. It follows that one cluster generally contains trajectories when the mice has larger bounding box overlaps or if the resident axis ratio is large. The other cluster thus contains trajectories where the mice bounding boxes do not overlap, and resident body is compact.

Figure 5: Learned programs on CalMS21. The subscripts represents the learned weights (in brackets) and biases (after the brackets) for the affine transformation followed by the bias.

| Model | CalMS21 | | | Basketball | | |
|---|---|---|---|---|---|---|
| | Purity | NMI | RI | Purity | NMI | RI |
| Random assignment | .597 | .000 | .536 | .500 | .000 | .500 |
| TVAE | .598 | .089 | .564 | .501 | .001 | .500 |
| TVAE+KMeans loss | .605 | .118 | .573 | .501 | .001 | .500 |
| JointVAE | .597 | .019 | .537 | .560 | **.034** | .507 |
| VQ-TVAE | .601 | .124 | .588 | .572 | .016 | .511 |
| Beta-TVAE | .616 | .115 | .589 | .565 | .013 | .509 |
| Ours (1 program) | .706 | **.423** | **.694** | **.596** | .027 | **.518** |
| Ours (2 programs) | .725 | .320 | .648 | .561 | .033 | .507 |
| Ours (3 programs) | **.756** | .314 | .633 | .584 | .022 | .514 |

Table 1: **Evaluating clusters from baseline and our neurosymbolic encoders on human-annotated labels.** Median purity, NMI, and RI on CalMS21 and Basketball compared to human-annotated labels (3 runs). Experiment hyperparameters are included in the appendix.

labels, and (2) average precision for behavior classification when integrating our programs into downstream tasks. For cluster metrics, we use Purity (Schütze et al., 2008), Normalized Mutual Information (NMI) (Zhang et al., 2006), and Rand Index (RI) (Rand, 1971). We report the median of three runs. More details, including the standard deviation and the ELBO, are in the appendix.

### 4.2 Q1: Are the clusters created with our programs meaningful?

**Synthetic dataset experiments.** Our synthetic dataset consists of trajectory data with 4 ground truth classes, corresponding to positive/negative x/y directions. The goal is to learn symbolic programs that capture the ground-truth classes, while leaving the neural latent space to capture any residual information, such as the random initial velocity. We visualize the 2 dimensions of the neural latent space of a TVAE along with 0, 1, and 2 learned programs in Figures 4c, 4d & 4e. The initial neural latent space of the TVAE contains 4 clusters corresponding to the 4 ground-truth classes in Figure 4c. After our algorithm learns the first program that thresholds the final x-position, the resulting latent space in Figure 4d captures the other factor of variation as 2 clusters corresponding to the final y-positions. Lastly, when our algorithm learns a second program that thresholds the final y-position, the resulting latent space in Figure 4e no longer contains any clear clustering, showing that our approach has successfully extracted the 4 ground-truth classes.

**Real-world datasets experiments.** We compare clusters produced by our neurosymbolic encoder with fully neural autoencoding baselines (Table 1), measured against human-annotated behaviors. For CalMS21,

| Model | CalMS21 (DSL 1) | | | CalMS21 (DSL 2) | | | CalMS21 (DSL 3) | | |
|---|---|---|---|---|---|---|---|---|---|
| | Purity | NMI | RI | Purity | NMI | RI | Purity | NMI | RI |
| Ours (1 program) | .706 | **.423** | **.694** | .689 | **.364** | **.681** | .649 | **.325** | .616 |
| Ours (2 programs) | **.725** | .320 | .648 | **.715** | .359 | .673 | **.664** | .324 | **.634** |

Table 2: **Effect of varying DSLs on CalMS21 for neurosymbolic encoders.** Median purity, NMI, and RI on CalMS21 of our algorithms with DSLs selected by three domain experts compared to human-annotated labels (3 runs). DSL1 corresponds to Table 1.

| Model | CalMS21 | | | Basketball | | |
|---|---|---|---|---|---|---|
| | Purity | NMI | RI | Purity | NMI | RI |
| TVAE | .598 | .089 | .564 | .501 | .001 | .500 |
| TVAE (w/ features) | .597 | .103 | .570 | .565 | .012 | .508 |
| VQ-TVAE | .601 | .124 | .588 | .571 | .016 | .511 |
| VQ-TVAE (w/ features) | .608 | .114 | .601 | .525 | .002 | .501 |
| Beta-TVAE | .616 | .115 | .589 | .566 | .013 | .509 |
| Beta-TVAE (w/ features) | .612 | .096 | .571 | .563 | .011 | .508 |

Table 3: **Effect of encoding DSL features into baselines.** Median purity, NMI, and RI on CalMS21 and Basketball compared to human-annotated labels (3 runs) for baseline with trajectory inputs only, and baseline with trajectory features added.

we observe that our method consistently outperforms the baselines in all three cluster metrics. We note that purity increases as the number of programs (thus clusters) increase, while NMI and RI decrease. This implies our method with two clusters best correspond to CalMS21 behaviors, but the other clusters found by our method may still be useful for domain experts. For Basketball, our method improves slightly with respect to purity, but is overall comparable with the baselines.

**Qualitative interpretation of our clusters.** We further study the programs and clusters produced by our algorithm for the CalMS21 dataset, through a qualitative study with a behavioral neuroscientist. Here, the behavioral neuroscientist analyzes the programmatic clusters produced from the symbolic representation of our neurosymbolic encoder for one, two, and three programs, resulting in two, four, and eight clusters respectively. The CalMS21 dataset is originally manually annotated with 4 classes corresponding to "attack", "investigation" (sniff), "mount", and "other" labels. "Other" corresponds to when no behaviors-of-interest is occuring, and is typically when the mice are not interacting.

In the single program case, our programs correspond to two discovered clusters. These clusters were classified by domain experts as referring to (1) when the mice are interacting and (2) when there are no interactions. They noted that this is based on distance between the mice, which is consistent with our program (Figure 5a) using distance between nose of resident and tail of intruder. For two programs, there are a total of four discovered clusters, with two clusters each corresponding to no interaction and interaction. For the interaction clusters, the domain expert was further able to identify sniff tail behavior as one of the clusters. In this case, the programs found were based on intruder head body angle, resident nose and intruder tail distance, and resident nose and intruder nose distance. The domain expert found the three program case to be more difficult to interpret, but was able to identify clusters corresponding to sniff tail, resident exploration, interaction facing the same direction (ex: mounting), and interaction facing opposite directions (ex: face-to-face sniffing).

### 4.3 Q2: How sensitive is our approach to the DSL?

**Choice of DSL.** To study the effect of DSL choices, we worked with three domain experts to construct three different DSLs used to learn our programmatic representations. These DSLs contained 8 to 10 different behavioral features for studying mouse social behavior on CalMS21, in addition to common sequential operations (Figure 3). A full list of features selected by domain experts are in the appendix.

While there is some variability, our approach consistently outperforms the baselines that contain fully neural encoders for all three DSLs (Table 2). Comparing some learned programs from two DSLs (Figures 5a, 5b), both contain a term that correlates with whether the mice are interacting (distance and bounding box overlap), and another term that correlates with resident speed (mice tends to be more stretched when they are moving quickly).

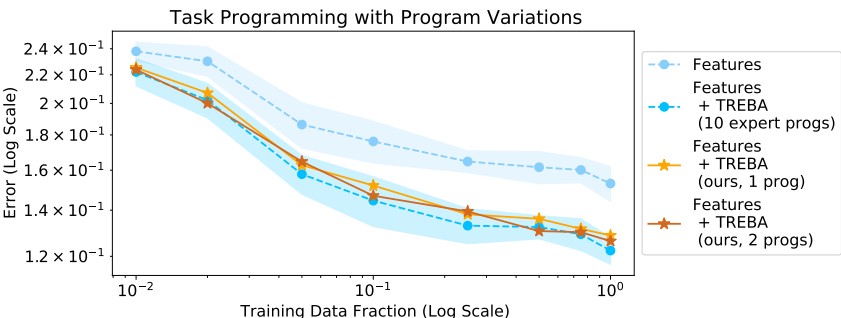

Figure 6: **Applying symbolic encoders for self-supervision.** "Features" is baseline w/o self-supervision. "TREBA" is a self-supervised approach in the Task Programming paradigm (Sun et al., 2021b), using either expert-crafted programs or our symbolic encoders as the weak-supervision rules. The shaded region is std dev over 9 repeats. The std dev for our approach (not shown) is comparable. Based on Sun et al. (2021b), the error is computed using $1.0 -$ Mean Average Precision.

**DSL features as input.** Lastly, we experiment with using the same DSL features introduced by domain experts as additional features for input trajectories instead (Table 3). For both CalMS21 and Basketball, the baselines using the additional features have comparable performance to using input trajectory data alone. In contrast, by using the features more explicitly as part of the DSL in our neurosymbolic encoders, we are able to produce clusters with a better separation between behavior classes based on cluster metrics (see Table 1).

### 4.4 Q3: Are the programs useful for downstream tasks?

We apply our programs to frame-level behavior classification (Segalin et al., 2020; Eyjolfsdottir et al., 2016; Burgos-Artizzu et al., 2012), where the goal is to automatically quantify behavior based on expert annotations. We are motivated by the observation that manual behavior annotation is time-consuming and expensive (Anderson & Perona, 2014), often being a bottleneck in the analysis workflow. Our unsupervised programs have the potential to reduce annotation effort and help accelerate behavioral studies, through the task programming framework. Task programming (Sun et al., 2021b) uses hand-crafted programs as self-supervision to improve behavior classification data efficiency; however, hand designing programs still requires human effort. Here, we show that unsupervised programs learned using our neurosymbolic encoders performs comparably to expert-designed programs on CalMS21.

We integrate the learned programs from our neurosymbolic encoder into the task programming framework (i.e., use them as a source of self-supervision instead of the expert-crafted programs), and compare to the classification performance using expert programs (Figure 6). The classification performance is computed using Mean Average Precision on the behaviors-of-interest in CalMS21 (attack, investigation, mount). Using only one program found using our approach, we are able to achieve comparable performance to 10 expert-written programs on the behavior classification task studied in Sun et al. (2021b). Importantly, we note that we automatically learned the self-supervision tasks from a DSL, instead of hand-crafting them as in Sun et al. (2021b). This demonstrates that programs found by our approach can be applied effectively to downstream behavior analysis tasks such as task programming.

## 5 Other Related Work

**Interpretable latent variable models.** Latent representations, especially those that are human-interpretable, can help us understand the structure of data. These models may learn disentangled factors (Higgins et al., 2016; Chen et al., 2016a; Ma et al., 2020) or semantically meaningful clusters (Ma et al., 2019) using unsupervised learning approaches. These approaches are often grounded in the VAE framework (Kingma & Welling, 2014). Some of these approaches, such as JointVAE (Dupont, 2018), Discrete VAE (Rolfe, 2016), Guided-VAE (Ding et al., 2020), and VQ-VAE (Oord et al., 2017), learn discrete latent representations (in particular, JointVAE learns a combined discrete-continuous representation, just like our approach). However, these approaches use fully neural encoders. To our knowledge, our work is the first to

propose neurosymbolic encoders, where the symbolic component produces a symbolic program that produces an low-dimensional encoding of the input data.

**Neurosymbolic programming.** Neurosymbolic programming (Chaudhuri et al., 2021) has seen much activity in the recent past. Existing approaches here are often trained in a supervised fashion (Gulwani, 2011; Wang et al., 2017; Shah et al., 2020; Cui & Zhu, 2021), or within a (generative) policy learning context with an explicit reward function (Chen et al., 2018; Verma et al., 2018; 2019; Bastani et al., 2018; Inala et al., 2020; Feinman & Lake, 2020; Trivedi et al., 2021). Prior work on unsupervised program synthesis has mostly addressed generative modeling, i.e., the synthesis of programs that can generate the training data (Tian et al., 2018; Ellis et al., 2018; Feinman & Lake, 2020). This task is analogous to learning a symbolic decoder rather than a symbolic encoder. Studying how to incorporate such methods into our framework can be an interesting future direction.

**Representation learning for behavior analysis.** Representation learning has been applied to a variety of downstream tasks for behavior analysis, such as discovering behavior motifs (Berman et al., 2014; Singh et al., 2021), identifying internal states (Calhoun et al., 2019), and improving sample-efficiency (Sun et al., 2021b). Studies in this area have used methods such as VAE (Kingma & Welling, 2014), AR-HMM (Wiltschko et al., 2015), forecasting or predicting future behaviors (Liang et al., 2020; Gao et al., 2020), and Umap (McInnes et al., 2018) to better understand the latent structure of behavior. Similar to a few other representation learning methods (Luxem et al., 2020; Sun et al., 2021b), we also use an encoder-decoder setup on trajectory data. However, our work learns a neurosymbolic encoder whereas existing works in this area have fully neural encoders. Our work can aid behavior analysis by learning more interpretable latent representations and can be applied to downstream tasks, such as behavior classification.

## 6 Discussion

We present a novel approach for unsupervised learning of neurosymbolic encoders. Our approach integrates the VAE framework with program synthesis and results in a learned representation with both neural and symbolic components. Experiments on trajectory data from behavior analysis demonstrate that our programmatic descriptions of the latent space result in more meaningful clusters relative to human-defined behaviors, compared to purely neural encoders. Additionally, we show the practicality of our approach by applying our learned programs to achieve comparable performance to expert-constructed tasks in a self-supervised learning approach for behavior classification.

**Problem Scope.** We explore unsupervised learning of neurosymbolic encoders for the first time, and here, our neurosymbolic encoders tackle domains consisting of lower dimensional spatiotemporal data. These types of domains covers a wide range of application areas, from behavioral data (animal behavior and sports analytics in our experiments), to control systems for rigid-body systems, to biomarkers or socioeconomic markers. In many of these domains, there are existing domain expertise that can be leveraged to create the DSL for our neurosymbolic encoders. For example, we use the behavioral features from Segalin et al. (2020) in our work. One direct application of learning semantically meaningful programs is that it can be used to improve learning pipelines, such as task programming, as we have demonstrated.

**Limitations.** One limitation of our current approach is scalability of the program search process. While our program search is parallelizable, such that learning additional programs would not incur significant additional time, the symbolic encoder update does increase the runtime over a purely neural solution. Here, we have explored our approach on settings where shorter programs are beneficial. Future work have the potential to further expand the applications of these models to larger, more complex systems. Furthermore, our approach requires programs that are differentiable with respect to its parameters. We note that there are increasingly more differentiable DSLs, such as Shah et al. (2020); Cui & Zhu (2021); Valkov et al. (2018); Gaunt et al. (2016); Bunel et al. (2016), and there are commonly-adopted ways to make differentiable approximations to more established non-differentiable DSLs (for example, in Shah et al. (2020), the authors use a smooth differentiable approximation of the non-differentiable if-then-else statement). These common challenges in using neurosymbolic learning in science is further discussed in Sun et al. (2022).

**Future Directions.** There are many future directions to explore for neurosymbolic encoders based on our work. Scalability is one important area as discussed above. Another direction is to extend this work to other domains such as image and text data, in order to learn interpretable symbolic latent representations. Neurosymbolic encoders on images would require a DSL for pixel data as well as architecture changes, such as using convolutional VAEs. Furthermore, one can improve upon our greedy approach in Algorithm 2 for finding the optimal set of symbolic programs, e.g. by performing local coordinate ascent in program space, similar to algorithms for large-scale neighborhood search (Ahuja et al., 2002). Lastly, while practically-oriented extensions of VAEs such as our own have yielded great practical benefit, they often lead to sub-optimal results from a pure likelihood (or ELBO) perspective. One final direction is to rigorously formulate a learning objective from the ground up that formally encapsulates practically-oriented extensions of VAEs.

**Acknowledgements.** The authors are grateful to the anonymous reviewers for their helpful comments. This work was funded in part by NSF #1918865, and a gift from Amazon.

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

## A    Additional Results

Table 4 contains the standard deviations of the results in Table 1 of the main paper.

Table 5 contains the median ELBO of our baselines and our neurosymbolic encoders. We find that our symbolic encoders are comparable with our baselines. This is expected: since we are imposing additional constraints on the encoder (a program with a bounded depth), we would not expect the variational approximation to be better than an encoder without these constraints (fully-neural encoder). In general, obtaining better or more semantically-meaningful cluster assignments can come at the cost of a smaller ELBO. For example, we find that introducing a clustering loss to the TVAE can result in better metrics, but in lower ELBO as well.

| Model | CalMS21 | | | Basketball | | |
|---|---|---|---|---|---|---|
| | Purity | NMI | RI | Purity | NMI | RI |
| TVAE | .002 | .011 | .001 | .049 | .012 | .008 |
| TVAE+KMeans loss | .001 | .002 | .001 | .006 | .001 | .001 |
| JointVAE | .000 | .003 | .022 | .037 | .020 | .004 |
| VQ-TVAE | .005 | .004 | .016 | .042 | .022 | .014 |
| Beta-TVAE | .001 | .001 | .001 | .124 | .140 | .088 |
| Ours (1 program) | .026 | .056 | .035 | .039 | .014 | .001 |
| Ours (2 programs) | .017 | .051 | .019 | .053 | .020 | .018 |
| Ours (3 programs) | .088 | .075 | .030 | .007 | .002 | .002 |

Table 4: Standard deviation of purity, NMI, and RI on CalMS21 and Basketball compared to human-annotated labels (3 runs). Random assignment metrics have standard deviation close to 0.

| Model | CalMS21 | Basketball |
|---|---|---|
| TVAE | 1120 | 895 |
| TVAE+KMeans loss | 1079 | 893 |
| JointVAE | 1090 | 902 |
| VQ-TVAE | 971 | 911 |
| Beta-TVAE | 1110 | 898 |
| Ours (1 program) | 1075 | 894 |
| Ours (2 programs) | 1073 | 893 |
| Ours (3 programs) | 1079 | 899 |

Table 5: Median ELBO of CalMS21 and Basketball across 3 runs.

## B    Implementation Details

The hyperparameters for our approach are in Tables 6,  7 and the hyperparameters for baselines are in Table 8. We used the Adam Kingma & Ba (2014) optimizer for all training runs. Specifically, Table 6 contains hyperparameters for program learning. Our use of the hyperparameters during the program learning process are the same as those from NEAR Shah et al. (2020). Table 7 contains the hyperparameters for training the VAE component of our model, including the hyperparameters we used for capacity.

| | n. epochs | s. epochs | frontier size | penalty | max depth | lr | batch size |
|---|---|---|---|---|---|---|---|
| Synthetic | 10 | 10 | 30 | 0.01 | 2 | 0.0002 | 32 |
| CalMS21 | 6 | 10 | 8 | 0.01 | 5 | 0.001 | 256 |
| Basketball | 8 | 8 | 30 | 0.01 | 3 | 0.002 | 128 |

Table 6: Hyperparameters for program learning. n. epochs and s. epochs represent the number of neural and symbolic epochs respectively, where the neural epoch is for the neural heuristic. lr is the learning rate.

| | epochs | z dim | h dim | RNN dim | adv. dim | disc. cap. | cont. cap. | lr |
|---|---|---|---|---|---|---|---|---|
| Synthetic | 50 | 4 | 16 | 16 | 8 | 0.6 | - | 0.0002 |
| CalMS21 | 30 | 8 | 256 | 256 | 8 | 0.69 | 10 | 0.0001 |
| Basketball | 20 | 8 | 128 | 128 | 8 | 0.6 | 4 | 0.02 |

Table 7: Hyperparameters for VAE training. The batch size is the same as the ones for program learning in Table 6.

| | JointVAE | | | VQ-TVAE | Beta-TVAE | | |
|---|---|---|---|---|---|---|---|
| | weight | disc. cap | cont. cap | # embeddings. | weight | cap | cap iter |
| CalMS21 | 100 | 0.69 | 10 | 4 | 100 | 20 | 10k |
| Basketball | 10 | 0.6 | 4 | 2 | 10 | 5 | 20k |

Table 8: Hyperparameters for baseline models. On CalMS21, the z dim for all baselines are 32 and trained for 200 epochs.

## B.1 Baseline Details

**TVAE.** We use a variation of the VAE where the inputs are trajectory data, called a TVAE Co-Reyes et al. (2018); Zhan et al. (2020); Sun et al. (2021b). Here, the neural encoder $q_\phi$ and decoder $p_\theta$ are instantiated with recurrent neural networks (RNN), where $\mathbf{z} \sim q_\phi(\cdot|\mathbf{x})$. In this domain, $\mathbf{x}$ is a trajectory of length $T$: $\mathbf{x} = \{x_1, \ldots, x_T\}$. The TVAE objective is:

$$\mathcal{L}^{\text{tvae}} = \mathbb{E}_{q_\phi}\left[\sum_{t=1}^{T} -\log(p_\theta(x_t|x_{<t}, \mathbf{z}))\right] + D_{KL}(q_\phi(\mathbf{z}|\mathbf{x})||p(\mathbf{z})). \tag{9}$$

All other baselines are variations of the TVAE, based on variations of VAE studied in recent works.

**TVAE + KMeans loss.** A few works Ma et al. (2019); Luxem et al. (2020) have studied adding a loss to the VAE framework to encourage clustering in the latent space, called the K-means loss. Given a data matrix $\mathbf{z} \in \mathbb{R}^{d \times N}$, the K-means objective is:

$$\mathcal{L}^{\text{k-means}} = Tr(\mathbf{z}^T\mathbf{z}) - Tr(\mathbf{A}^T\mathbf{z}^T\mathbf{z}\mathbf{A}), \tag{10}$$

where $\mathbf{A} \in \mathbb{R}^{N \times k}$ is called the cluster indicator matrix. We optimize this loss using the implementation in Luxem et al. (2020), where $\mathbf{A}$ is updated by computing the $k$-first singular values of $\sqrt{\mathbf{z}^t\mathbf{z}}$. The K-means loss is trained jointly with the TVAE loss (Eq 9) as one of our baselines.

**JointVAE.** JointVAE Dupont (2018) is a variation of VAE that jointly optimizes discrete ($\mathbf{c}$) and continuous ($\mathbf{z}$) latent variables. The JointVAE objective encourages the KL divergence terms to match capacities $C_z$ and $C_c$ that gradually increases during training. The objective is:

$$\mathcal{L}^{\text{jointvae}} = \mathbb{E}_{q_\phi}[\log p_\theta(\mathbf{x}|\mathbf{z}, \mathbf{c})] - \gamma|D_{KL}(q_\phi(\mathbf{z}|\mathbf{x})||p(\mathbf{z})) - C_z| - \gamma|D_{KL}(q_\phi(\mathbf{c}|\mathbf{x})||p(\mathbf{c})) - C_c|, \tag{11}$$

where $\gamma$ is a constant. Since the capacities of the discrete and continuous variables are controlled separately, the model is forced to encode information using both channels. Here, we use the trajectory formulation of JointVAE, where:

$$\log p_\theta(\mathbf{x}|\mathbf{z}, \mathbf{c}) = \sum_{t=1}^{T} \log p_\theta(x_t|x_{<t}, \mathbf{z}, \mathbf{c}). \tag{12}$$

**VQ-TVAE.** VQ-VAE Oord et al. (2017) combines vector quantization with VAEs. These models produce discrete latent encodings that are used to index an embedding table (or codebook). $\mathbf{z}_e$ , the continuous output of the encoder, is mapped to a discrete encoding based on its nearest neighbor in the codebook, then the indexed encoding $\mathbf{z}_q$ is used as input to the decoder. During training, the model learns the codebook, as well as the assignments. The objective is:

$$\mathcal{L}^{\text{vqvae}} = \log p_\theta(\mathbf{x}|\mathbf{z}_q) + ||\text{sg}[\mathbf{z}_e] - e||_2^2 + \beta||\mathbf{z}_e - \text{sg}[e]||_2^2, \tag{13}$$

where $e$ are embeddings from the codebook, and sg represents the stopgradient operator.

**Beta-TVAE.** Beta-VAEs Higgins et al. (2016); Burgess et al. (2017) have been shown to learn disentangled representations from the image domain. As originally proposed, an adjustable hyperparameter $\beta$ is used to weigh the KL term in the VAE objective. We use the version of beta-vae training objective with gradually increasing capacity $C$ proposed in Burgess et al. (2017). This object is:

$$\mathcal{L}^{\text{betavae}} = \mathbb{E}_{q_\phi}[\log p_\theta(\mathbf{x}|\mathbf{z})] - \gamma |D_{KL}(q_\phi(\mathbf{z}|\mathbf{x})||p(\mathbf{z})) - C|, \tag{14}$$

where $\gamma$ is a constant. Here, we apply the beta-VAE objective to trajectory data using the factorization shown in Eq 8.

## B.2 Metrics Definition

We evaluate our programs quantitatively using standard cluster metrics relative to human-defined labels. The metrics we use are Purity (Schütze et al., 2008), Normalized Mutual Information (NMI) (Zhang et al., 2006), and Rand Index (RI) (Rand, 1971). These metrics have also been used by other works for evaluating clustering (Ma et al., 2019; Luxem et al., 2020). The definition of purity is:

$$Purity = \frac{1}{n} \sum_{u \in U} \max_{v \in V} |u \cap v| \tag{15}$$

where $U$ is the set of human-defined labels, $V$ is the set of cluster assignments from the algorithm, and $n$ is the total number of trajectories.

The NMI is defined as:

$$NMI = \frac{\sum_{u \in U} \sum_{v \in V} |u \cap v| \log \left( \frac{n|u \cap v|}{|u||v|} \right)}{\sqrt{\sum_{u \in U} |u| \log \frac{|u|}{n} \sum_{v \in U} |v| \log \frac{|v|}{n}}} \tag{16}$$

RI is defined as:

$$RI = \frac{TP + TN}{n(n-1)/2} \tag{17}$$

where $TP$ are the number of trajectory pairs correctly placed into the same cluster, $TN$ are the number of trajectory pairs correctly placed into different clusters, and $n$ is the total number of trajectories. For all metrics, a value closer to 1 indicates clusters that more closely match the human-defined labels.

## C Dataset and DSL Details

**Synthetic.** We generate trajectories with the following steps:

1. Sample initial position $x_1 \sim \mathcal{N}([10, 10], [1, 1])$.

2. Sample velocity from $v = [v_x, v_y] \sim \mathcal{N}([0, 0], [1, 1])$ such that $0.05 < \|v\|_2 < 0.4$.

3. Sample force in $x$-direction $c_x \sim \text{Bernoulli}(0.5)$ and update $v'_x = v_x + 0.4 \cdot (2c_x - 1)$.

4. Sample force in $y$-direction $c_y \sim \text{Bernoulli}(0.5)$ and update $v'_y = v_y + 0.4 \cdot (2c_y - 1)$.

5. Generate trajectory with $x_{t+1} = x_t + v' + 0.2 \cdot \epsilon_t$, where $\epsilon_t \sim \mathcal{N}(0, 1)$.

$v'$ is fixed for an entire trajectory. $(c_x, c_y)$ defines a label for each trajectory (one of 4). The ground-truth decoder is linear with respect to $x, v, c_x, c_y$. The DSL for the synthetic dataset includes library functions that threshold the final $x$ and $y$ positions, used to demonstrate that the ground-truth can be learned and the information can be extracted from the neural latent space (Figure 4). Experiments were run locally with an Intel 3.6-GHz i7-7700 CPU with 4 cores and an NVIDIA GTX 1080 Ti GPU with 3584 CUDA cores.

**CalMS21.** The CalMS21 dataset Sun et al. (2021a) consists of trajectory data from a mouse tracker Segalin et al. (2020), where each mouse is tracked by seven body keypoints from an overhead camera. The two mice are engaging in social interaction, where an intruder mouse is introduced to the cage of the resident mouse. The dataset contains an unlabelled split which we use for training and validation, and we use the test split of

Task 1 in CalMS21 for testing. Each frame of the test split is annotated by a domain expert with one of four labels: attack, mount, investigation, other. We use these annotated behavior labels for comparison with clusters produced by our algorithm. This dataset is available under the CC-BY-NC-SA license.

The feature selects in the CalMS21 DSL are based on behavior attributes computed on trajectory data from domain experts in this area Segalin et al. (2020). In particular, we asked three domain experts to independently select features from Segalin et al. (2020) to be part of the DSL. The time it takes domain experts to do this step is on the timescale of minutes. A full list of all features use in the DSLs are as follows:

- Features in DSL 1: head body angle (resident and intruder), social angle (resident and intruder), speed (resident and intruder), distance between nose of resident and tail of intruder, distance between nose of resident and nose of intruder.

- Features in DSL 2: distance between head of mice, distance between body of mice, distance between head of resident to body of intruder, resident acceleration, resident nose speed, resident axis ratio of fitted ellipse, intersection over union of mice bounding boxes, resident social angle, distance between nose of resident and tail of intruder, distance between nose of resident and nose of intruder.

- Features in DSL 3: head body angle (resident and intruder), area of ellipse fitted to body keypoints (resident and intruder), acceleration (resident and intruder), distance between nose of resident and tail of intruder, distance between nose of resident and nose of intruder.

Note that unless otherwise stated, the CalMS21 experiments uses the features from DSL 1.

The experiments are ran on Amazon EC2 with an Intel 2.3 GHz Xeon CPU with 4 cores equipped with a NVIDIA Tesla M60 GPUs with 2048 CUDA cores.

**Basketball.** The basketball dataset was also used in Shah et al. (2020); Zhan et al. (2020) and tracks the $xy$-positions of players from real NBA games. The positions are centered on the left half-court. Both (5) offensive and (5) defensive players are tracked, as well as the ball (excluded in our experiments).

The DSL for basketball contains library functions that compute the speed, acceleration, final positions, and distance-to-basket of players and take the maximum, minimum, or average over the players. We did not consult a domain expert for this DSL, but these functions were used as labeling functions in Zhan et al. (2020). Basketball experiments were run locally with an Intel 3.6-GHz i7-7700 CPU with 4 cores and an NVIDIA GTX 1080 Ti GPU with 3584 CUDA cores.

