# OpenReview forum: "Unsupervised Learning of Neurosymbolic Encoders"
_TMLR — Accepted by TMLR_

### Review · Reviewer_Hvn1 · 2022-08-15

**Summary Of Contributions:**

This paper proposes neurosymbolic encoders, where the encoder contains both a pure neural component and a symbolic component. The symbolic encoder learns latent representation as symbolic programs defined in domain-specific languages. To train the neurosymbolic encoder, they initiate the encoder with a pure neural one; then in each iteration, they greedily select a grammar rule in their DSL to replace a nonterminal, so that it remains similar to the current latent representation. They integrate the neurosymbolic encoder into the VAE framework for unsupervised learning. They evaluate their approach on a synthetic benchmark, CalMS21 benchmark containing trajectories of mice behavior, and a basketball benchmark containing trajectories of basketball players. They demonstrate that their neurosymbolic encoder achieves better clustering performance, and the learned programs have comparable performance to expert-crafted programs on behavior classification, when they are integrated into the task programming framework.

**Broader Impact Concerns:**

No ethical concerns.

**Requested Changes:**

1. Please provide a comparison of training time between neurosymbolic encoders and baselines.

2. Please provide more discussion on the effect of different number of programs for downstream tasks. For example, in Figure 5, will 10 learned programs outperform expert-crafted programs?

3. Please provide more discussion on the scalability and applicability of the approach; i.e., in which scenarios and problem scopes that the approach will be beneficial.

**Strengths And Weaknesses:**

Strengths

1. To my knowledge, this is the first work proposing to learn neurosymbolic encoders in the unsupervised setting. This is a promising direction for learning interpretable representations.

2. The authors demonstrate good results on several behavior analysis benchmarks. In particular, the learned programs perform comparably to expert-annotated programs for task programming.

Weaknesses

My main concern of this work is about the efficiency, scalability and applicability of the proposed approach.

1. How does the training time of neurosymbolic encoders compare to standard neural encoders? I feel that the update of the symbolic encoder imposes a large overhead.

2. For classification tasks, does increasing the number of programs reduce the error? For example, in Figure 5, will 10 learned programs outperform expert-crafted programs?

3. While the neurosymbolic encoder outperforms standard encoders on CalMS21, there is no much gain on Basketball benchmarks. Although the framework has potential to extend to different domains, currently it is unclear in which scenarios the approach would help. For more complicated learning problems, I think more programs and more complicated DSLs are needed, and it will require more scalable training algorithms.

---

> ### Author Response · Authors · 2022-09-02
> **Response to Reviewer**
>
> We really appreciate the reviewer for their helpful comments and suggestions! We have uploaded a revision of the paper with the reviewer's suggestions. More detailed response below.
>
> Response to requested changes and weaknesses (we respond to corresponding changes and weaknesses together):
>
> > "Please provide a comparison of training time between neurosymbolic encoders and baselines."
>
> Our neurosymbolic encoder takes longer to train compared to the baseline. For example, on CalMS21, the standard TVAE baseline trains in ~9 hours, while our method with 1 program trains in ~24 hours. Scalability is certainly a concern and arguably the most important direction for future work.  The goal of this paper is the first interesting non-trivial demonstration of unsupervised learning of neurosymbolic encoders. Note that our program search is parallelizable, and with more machines, learning additional programs would not incur significant additional time.
>
> > "Please provide more discussion on the effect of different number of programs for downstream tasks."
>
> We don’t expect larger amounts of learned programs to have a significant effect on performance, since (1) we see only a small improvement when going from 1 to 2 learned programs in our setting and (2) the original paper on task programming (Sun et al., 2021b) found that 3 expert-crafted programs performed only slightly worse than using 10 programs.
>
> > "Please provide more discussion on the scalability and applicability of the approach"
>
> The problem scope we tackle is lower dimensional spatiotemporal data where we have domain expertise (ex: control systems for rigid-body systems, biomarkers or socioeconomic markers, behavioral tracking). One obvious useful application of learning semantically meaningful programs is that it can be used to improve learning pipelines, such as task programming as we demonstrated in our paper. Our goal with this paper is to demonstrate unsupervised learning of neurosymbolic encoders in non-trivial settings for the first time. We agree with the reviewer on scalability, and see our earlier response in the runtime comment. We have also updated the conclusion/discussion in Section 6 with these points from the reviewer.
>
> We thank the reviewer again for these helpful comments and we have uploaded a new paper revision.

---

### Review · Reviewer_BbGj · 2022-08-18

**Summary Of Contributions:**

The authors propose representing data using a representation that has both a neural and a symbolic component. To obtain the symbolic component they closely follow the implementations of Shah et al. . The authors demonstrate their model initially on a toy dataset. They show how the neural part of the model learns factors of variation that the symbolic part of the model has not learned.

Impressively, they show results on two real world datasets: Mouse trajectories and basketball. They show good clustering performance and use domain experts to classify the clusters.

Finally, they demonstrate their programs being used in down steam tasks. Showing similar performance to expert crafted programs.

**Requested Changes:**

1. Please improve the clarity of the methods section
 - What are the dimensions of z_{a, \phi}?
 - How is z_{a, \phi} obtained from q(a, \phi)
 - Reduce dependancy on Shah et al.
 - How are the losses from Section 3.3 combined?
 - Provide an improved description of the CaLM21 task.

2. Improve clarity of the "Qualitative interpretation of our clusters” section.

3. Report accuracy on both the down-steam CaLM21 and Basketball tasks.

**Strengths And Weaknesses:**

Strengths:
- Results on the synthetic data is very convincing and demonstrates well how the model works.
- Authors show results on real world datasets and demonstrate that their model outperforms (CaLMS21) or performs as well as baselines (Basketball).
- When learning to solve down-steam tasks, authors also show that their learned programs give a similar boost to performance as 10 hand-crafted ones.

Weaknesses:
- There is significant dependancy on domain expertise to define the DSL's and appears to only work for low dimensional data with well defined features.
- The methods section of the paper is very unclear and assumes a deep understanding of Shah et al.
- It is hard to interpret the results on down stream tasks because (a) it is not clear which dataset is used and (b) classification performance is not reported only error.
- Part of the motivation for this work is that the symbolic programs are more interpretable, however, the authors note that "The domain expert found the three program case to be more difficult to interpret".


General comments and questions:

1. What do the authors mean by “our approach offers significantly better separation”?
2. The following statement is vague, “these representations can potentially be more factorized or well-separated”
3. What is the relation to inverse graphics? e.g. [1], [2].
4. Section 2.2 assumes comprehensive knowledge of Shah et al. and this section is hard to understand without first reasoning Shah et al. first.
5. Why is phi not fixed when optimising Eqn 3? Is this because a’ may require different parameters?
6. Thought Figure 4 does convey the essence of the learned programs, its not clear what the individual functions are or what form they take.
7. In “Synthetic dataset experiments” should the reference be to Figures c-e rather than b-d?
8. Figure 3 is very interesting. It’s very nice to see that the neural part of the representation really does only learn what the symbolic part has not learned. This is very impressive.
9. How much do the results in Figure 3 depend on the regularisation described in Section 3.3?
10. It is not clear how the regularisers in Section 3.3 are incorporated? Are the extra terms simply added? Is there a co-efficient. It would be nice to see this in the appendix and a brief discussion in the main text.
11. In Figure 3(b) can you please explain the threshold values? Why are these values not closer to zero? Would that not be the optimal solution?
12. In “Synthetic dataset experiments” please make clear that TVAE refers to the neural component of the representation.
13. DSL 1 and DSL 2 are not introduced in the main text before being referenced in Figure 4. It would be helpful to point to the appendix (possibly when introducing Q2 in Section 4).
14. To aid interpretation of results on the CalMS21 dataset (Figure 4) it would be helpful to list the four behaviours of the mice. Without referring to the Appendix it is very hard to interpret the results in Figure 4. The interpretation of the program in 4(a) also only makes sense in the context of understanding what the four behaviours are.
15. For all Tables please highlight the best result in bold.
16.  To be clear in "Qualitative interpretation of our clusters” are experts analysing the neural clusters? Please make this clear. It could be helpful to show these clusters (possibly in the appendix) along with some figures showing the examples of trajectories which domain experts consider typical for the cluster.
17. “In the single program case, the domain expert classified the discovered clusters as when the mice are interacting and when there are no interaction” This sentence is hard to parse. I would suggest something of the form. “In the single program case our model discovers two clusters. These clusters were classified by domain experts as referring to …”
18. Why is is that case that 1 program results in two clusters while 2 programs result in 4?
19. It is not clear to me how you obtain z_{a, \phi} from q(a, \phi). Is it just the output of the program (a, \phi)? What are its dimensions?
20. In Section 4.4 what is meant by frame level behaviour?
21. Down steam tasks: It’s very encouraging that a few learned programs performs as well as 10 hand-crafted programs.
22. Is Figure 5 reporting test error? Why not show classification performance?
23. Which dataset is used to obtain results in Figure 5? Why are there not results for both CaLMS21 and Basketball?
24. It's interesting in Table 1 that scores tend to be better for a single program than for three programs. Could the authors please comment further on why this may be the case?

---

> ### Author Response · Authors · 2022-09-02
> **Response to Review (1/2)**
>
> We would like to thank the reviewer for all the helpful comments and suggestions! We have updated a revision of the paper addressing many of the changes. A more detailed response is below.
>
> Response to Weaknesses:
>
> > "significant dependancy on domain expertise"
>
> The dependency on the DSL is inherent to almost all program learning approaches, and in general is both a strength and a weakness of program learning (which our approach inherits).  The weakness is the need to define a DSL; the strength is a much stronger (and sometimes more interpretable) inductive bias. Furthermore, in many domains, one can easily use existing domain expertise to quickly create a DSL at minimal extra work for the domain expert.  For example, features in the mouse behavior DSL already exist in the domain, from existing works (Segalin et al., 2020). Relative to, say, images, such data is comparatively low-dimensional, but arises in many domains that use spatiotemporal data (e.g., biomarkers such as EKGs, behavior tracking, control of rigid-body systems).
>
> > "methods section"
>
> We originally had additional details in Appendix D, which we have now moved into Section 3.2. This does increase the length of the paper, and so if the reviewers prefer the shorter version we can revert the change. We have also updated the methods section with the suggested changes from the reviewer.
>
> > "hard to interpret the results on down stream tasks"
>
> We apologize for the confusion, and have clarified in the text that the downstream task is from CalMS21.  Our error plot is consistent with the way that task programming (baseline) measures results, and is directly computed from classification performance (error = 1 - Mean Average Precision).
>
> > "the three program case"
>
> To be clear, the domain expert found the 8-cluster (=3 programs) neurosymbolic model more difficult to interpret than the 2-cluster and 4-cluster neurosymbolic models, which can be thought of as stress testing the approach.  Note also that our neurosymbolic models are generally much more interpretable that using a purely deep learning alternative (which do not result in any interpretable programs).
>
> Response to General Comments (we respond to all questions that have not been addressed by either the weaknesses or the requested changes):
>
> > Q1
>
> We meant separation of meaningful categories/clusters in the data (ex: such as behaviors-of-interest), as measured by standard unsupervised learning metrics (with respect to human-annotated behaviors), compared to our discovered clusters. We will update the Abstract so this is more clear.
>
> > Q2
>
> Similarly to point (1), we have updated the Introduction.
>
> > Q3
>
> We are happy to discuss this, but could the reviewer please link us to the references they refer to in [1],[2]? We do not see them in the comments. Generally, in inverse graphics, the decoder is usually fixed and symbolic (i.e., it is a renderer) and the goal is to learn a symbolic encoding of the raw data (e.g., shape, appearance, etc) that can be rendered.  In our work, we are actually learning a decoder (which right now is purely neural), and it would be interesting to combine all concepts into a single framework, where the domain expert can pick and choose what is fixed vs learned, what is symbolic vs neural.
>
> > Q4
>
> Added Section 3.2 (previously appendix D) that includes more details about Shah et al.
>
> > Q5
>
> Phi isn’t in Eqn 3; for the Eqn 3 variables, psi is fixed, but psi’ is optimized because yes, we still need to optimize the parameters of alpha’.
>
> > Q6
>
> The individual functions are based on behavioral features from Segalin et al. 2020 and the programs learn the affine transformation over the subset of specific features. For more description on the individual features, Table 3 in Segalin et al. 2020 has more details. If the reviewer feels these feature descriptions further help with understanding the program, we are happy to add these more detailed descriptions to the Appendix.
>
> > Q7
>
> Yes! Updated bcd to cde.
>
> > Q9
>
> We found that the regularizations were crucial to the approach. Both index and posterior collapse are quite common, especially while attempting to learn discrete latent variables. We borrowed these techniques developed in previous work, and added a few sentences in Section 3.4 of the main text to describe the phenomena.
>
> > Q10
>
> Yes they are added, and we also added this to the main text as part of Eq. 7.
>
> > Q11
>
> The ground-truth values are (10, 10), which corresponds to the mean starting position (listed in appendix). The threshold values may seem far from the ground-truth, but the programs actually capture the ground-truth quite well. This is because the synthetic dataset is generated in a way that final positions look like a symmetric bimodal Guassian distribution with minimal mass near the thresholds (somewhat depicted in Figure 4(a)).

---

> > ### Author Response · Authors · 2022-09-02
> > **Response to Review (2/2)**
> >
> > > Q12
> >
> > We modified Section 4.2 so this is more clear.
> >
> > >Q13
> >
> > We modified the Q2 introduction.
> >
> > >Q14
> >
> > We added the behaviors to Section 4.1.1.
> >
> > >Q15
> >
> > We highlighted the best results in bold.
> >
> > >Q18
> >
> > Our algorithm results in 2^k classes where k is the number of learned programs (see Section 3.3). This is a modeling choice, as we find binary programs to be the most interpretable.
> >
> > >Q19
> >
> > The dimensions of $z_{a, \phi}$ are equal to the number of classes the program intends to classify (2 for binary programs). $z_{a, \phi}$ is sampled using a Gumbel-Softmax (essentially the reparameterization trick for discrete variables). We clarify this in Section 3.3.
> >
> > >Q20
> >
> > Frame-level behaviors are behaviors that are annotated at each frame (instead of each video). All the CalMS21 behaviors are annotated at each frame. We updated the CalMS21 description in Section 4.1.1 to clarify this.
> >
> > >Q22
> >
> > The error is computed directly from classification performance, using 1.0 - Mean Average Precision, same as task programming (baseline).
> >
> > >Q23
> >
> > Addressed in requested changes.
> >
> > >Q24
> >
> > This is likely because 3 programs correspond to 8 clusters, and this may be over-segmenting the data; for comparison, CalMS21 has 4 human-annotated classes and Basketball has 2 classes. Table 1 compares discovered clusters to existing classes using standard clustering metrics, measuring shared information between the two sets of clusters (Sections 4.1.2, and details in Appendix B2). The results suggest that in these datasets, adding more clusters does not produce more informative clusters relative to existing human-annotated labels.
> >
> > Response to Requested Changes:
> >
> > > "dimensions of z_{a, \phi}"
> >
> > The dimensions of z_{a, \phi} are equal to the number of classes the program intends to classify (2 for binary programs), clarified in Section 3.3.
> >
> > > "How is z_{a, \phi} obtained from q(a, \phi)"
> >
> > z_{a, \phi} is sampled using a Gumbel-Softmax (essentially the reparameterization trick for discrete variables), clarified in Section 3.3.
> >
> > > "Reduce dependancy on Shah et al."
> >
> > We have added a new Section 3.2 (previously Appendix D) with details on NEAR (Shah et al. 2020).
> >
> > > "losses from Section 3.3"
> >
> > We’ve added a new Eq. 7 to make this more clear.
> >
> > > " improved description of the CaLM21 task"
> >
> > We have updated Section 4.1.1 with more details on the CalMS21 dataset and behaviors.
> >
> > > "Qualitative interpretation of our clusters"
> >
> > We've updated the qualitative interpretation paragraph as suggested.
> >
> > > "down-steam CaLM21 and Basketball tasks"
> >
> > The Task Programming framework is designed for animal behavior on CalMS21, and so there is no existing baseline using expert programs on Basketball. Thus, we chose to demonstrate the performance of downstream tasks on CalMS21.
> >
> > We thank the reviewer again for their helpful comments and feedback and have uploaded a revision of the paper with these suggestions.

---

> ### Comment · Reviewer_BbGj · 2022-09-02
> **References [1] and [2]**
>
> Apologies for not including these earlier.
> [1] Synthesizing Programs for Images using Reinforced Adversarial Learning
> [2] Deep Convolutional Inverse Graphics Network

---

> > ### Author Response · Authors · 2022-09-06
> > **Additional Response**
> >
> > Thanks for including the new references! [1] is complementary to our work in that their decoder is fixed and symbolic, while our decoder is neural and learned. See our discussion above in Q3. [2] also trains a neural decoder, but uses a specific sampling scheme to learn known factors of variation in the symbolic encoding. In contrast, in our work, these factors are not known ahead of time, and is learned by our program synthesis approach.

---

> ### Comment · Reviewer_BbGj · 2022-09-07
> **Response to rebuttal**
>
> 1. Thanks for adding eqn 7. it's now much more clear how you combine the losses.
> 2. The implementation of z_a_\phi is also more clear now, thank you.
> 3. Section 3.2 is a great addition and makes the paper more self contained.
> 4. Results in Figure 6 would be more clear presented as MAP. However, the caption is clear and the results are impressive.

---

### Review · Reviewer_rHaE · 2022-08-23

**Summary Of Contributions:**

The paper proposes a method to learn a neurosymbolic encoder within a variational autoencoder framework. The authors instantiate the symbolic portion of the encoders as (potentially incomplete) programs in a functional domain-specific language, following the approach in Shah et al 2020. These programs can have unexpanded nonterminals; the computations for these holes are done using neural networks instead. The proposed algorithm trains the encoder using a coordinate descent procedure where the depth of the program is extended in each step (thus reducing the role of the unexpanded nonterminals). The paper demonstrates the value of the method with experiments on trajectory datasets, showing that it creates more meaningful clusters compared to baselines, works with different DSLs, and is useful for the downstream application of Task Programming.

**Broader Impact Concerns:**

I have no particular concerns about broader impact. There may be objectionable applications made possible by this work, but its focus is not on any such applications, as is with most other machine learning papers.

**Requested Changes:**

Critical changes:
- Explain further about the impacts of adversarial information factorization and channel capacticy constraint, for example by providing an ablation study where they are not used.
- The work is largely about creating symbolic encoders, but there is also an unstructured neural encoder used alongside it.
  - What happens if you don't have these latent variables from the neural encoder?
  - In the channel capacity constraint, how are $\gamma_\phi$, $C_\phi$, $\gamma_{(\alpha, \phi)}$, and $C_{(\alpha, \phi)}$ set?

Not critical:
- In Section 3.1, there is a reference to $\alpha_0$ of Section 2.2, but Section 2.2 doesn't explicitly mention $\alpha_0$.
- In the caption of Figure 2, "different approximation" should be "differentiable approximation"
- Various citations should use `\citep` but do not, for example directly below equation 6. Others use `\citep` instead of `\citet`, for example at the end of the second-to-last paragraph of page 6.
- It is not completely clear how equations 4 and 5 are used to augment equation 2. It would be better to show the complete augmented equation separately.



**Strengths And Weaknesses:**

Strengths
- The work provides an interesting extension/application of the prior work Shah et al 2020 by applying it to the variational autoencoder setting.
- The paper is honest about limitations in the experimental results.
- The usefulness of the method is shown through multiple evaluation methods.

Weaknesses
- There is no longer a graph search, unlike Shah et al. 2020, and instead the program appears to be learned in a more greedy manner.
- DSL requires full differentiability of all operations, so it may be hard to extend.
- The stated algorithm requires many rounds of training to produce the symbolic encoder, and thus may be less practical for larger or more complicated datasets.
- The generated programs may not necessarily be particularly interpretable, as there is no constraint/cost to encourage interpretability.

---

> ### Author Response · Authors · 2022-09-02
> **Response to Review**
>
> We really appreciate the helpful comments from the reviewer! We've updated the paper with suggestions from the reviewer and see our detailed response below.
>
> Response to weaknesses:
>
> > "no longer a graph search"
>
> The main place we’re currently leveraging Shah et al., 2020 is the formulation of neural relaxations on non-terminal nodes, which can also be applied to more greedy search methods.  This aspect of Shah et al. 2020 is arguably the most useful in terms of enabling any method to work.
>
> > "full differentiability"
>
> This is an important point, although perhaps not as big a roadblock as one might expect.  There are increasingly more differentiable DSLs (NEAR (Shah et al., 2020), dPads [A], HOUDINI [B], TerpreT [C], ANC [D]), and there are commonly-adopted ways to make differentiable approximations to more established non-differentiable DSLs (for example, in Shah et. al 2020, the authors use a smooth differentiable approximation of the non-differentiable if-then-else statement).
>
> > "rounds of training"
>
> Scalability is certainly a concern and arguably the most important direction for future work. The goal of this paper is the first interesting non-trivial demonstration of unsupervised learning of neurosymbolic encoders. Notably, our program search is parallelizable, and with more machines, learning additional programs would not incur significant additional time requirements. We have added more discussion on this in Section 6.
>
> > "no constraint/cost to encourage interpretability"
>
> In almost all program learning approaches, interpretability is encouraged through a combination of good DSL design and having a structural cost to encourage “simpler” programs.  Our approach also adopts this paradigm, and we do have a structural cost (similar to Shah et al, 2020), see Eq. 4. For behavior analysis, interpretability of similar programs (learned in a supervised way) compared to fully neural solutions has been studied in [E]. Here, we study these programs in an unsupervised setting for the first time.
>
> [A] Cui & Zhu, Differentiable Synthesis of Program Architectures, NeurIPS 2021
>
> [B] Valkov et al., HOUDINI: Lifelong Learning as Program Synthesis, NeurIPS 2018
>
> [C] Gaunt et al., Terpret: A probabilistic programming language for program induction
>
> [D] Bunel & Desmaison  et al., Adaptive Neural Compilation
>
> [E] Tjandrasuwita et al., Interpreting Expert Annotation Differences in Animal Behavior, CV4Animals at CVPR 2021
>
>
> Response to requested changes:
>
> > "impacts of adversarial information factorization and channel capacticy constraint"
>
> The impact of these additional losses have been demonstrated in prior work (adversarial information factorization in Creswell et al., 2017, and channel capacity constraint in Dupont, 2018). In our work, we use them similarly to previous work to address index collapse and posterior collapse respectively, and we have added a description of these modes of collapse if these losses are not used in Section 3.4 of the main text.
>
> > "What happens if you don't have these latent variables from the neural encoder?"
>
> Our synthetic experiment showcases the need to have a neural encoder alongside a symbolic one, as there exists additional ground-truth factors of variation (velocity) not captured by the learned programs. We have clarified this in Section 4.2. In this case, without the neural encoder, the decoder would not have enough information from the symbolic encoder to perform the reconstruction.
>
> > "channel capacity constraint"
>
> These are hyperparameters (Appendix Table 7). They are based on the channel capacity work from (Dupont, 2018).
>
> We have also fixed all the non-critical changes pointed out by the reviewer, notably the "complete augmented equation" is added as Equation 7 in the new paper revision. We would like to thank the reviewer again for the helpful feedback.

---

### Author Response · Authors · 2022-09-02
**Paper Revision Update**

We thank all reviewers for the helpful comments and feedback! We have responded to each reviewer in individual comments as well as uploaded a revision of the paper with the following changes:

*    Updated Section 3 on Methods, notably Section 3.2 describing NEAR (previous Appendix Section D), Section 3.4 describing index and posterior collapse with Eq 7 added, as well as some clarifications to text and notation.
*    Updated Section 4 on Experiments, notably Section 4.2 to describe role of neural encoder in synthetic experiments and Qualitative Interpretation paragraph, as well as added more details to the description of CalMS21 in Section 4.1.1.
*    Updated conclusion (renamed to discussion) with comments on scalability, applicability, and problem scope.

Minor Changes:
*    Fixed citation formatting
*    Some text updates:
     *    Updated abstract: “significantly better separation” → “significantly better separation of meaningful categories”
     *    Updated intro: “these representations can potentially be more factorized or well-separated” → “these representations can potentially be more factorized or well-separated into meaningful categories”
*    Added DSL1 and DSL2 reference to Q2 introduction
*    Highlighted best results in tables where comparison is important

Please let us know if you have any additional comments - thanks!

---

### Decision · Action_Editors · 2022-10-06

**Recommendation:** Accept with minor revision

**Comment:**

This paper presents an approach for learning neural symbolic encoders that can encode data into programmatic latent representations within domain specific languages.  The proposed approach learns these encoders in an unsupervised way, through a VAE framework.  The work is an extension of Shah et al., 2020 with a few innovations, most notably the VAE framework and the corresponding learning algorithms.

All reviewers are leaning accept after the discussion, hence I’m also recommending an accept.

There are however a few common concerns raised by the reviewers that the authors can consider addressing further, in particular the programs being generated are small and toy-ish in this paper, which may have its place in some applications but scaling up to larger more complex programs with more complicated control flow seems to be a big challenge.  Also, the approach requires the program semantics to be fully differentiable with respect to the parameters, which also seems to be strong limitation.

The paper presents some discussions in Section 6, but could benefit from discussing some of the limitations a bit more, a journal paper is a good place to do this as you have more pages to work with.

**Audience:**

Yes.

**Claims And Evidence:**

Yes.